# Assessment of the Toxic Metals Pollution of Soil and Sediment in Zarafshon Valley, Northwest Tajikistan (Part II)

**DOI:** 10.3390/toxics8040113

**Published:** 2020-11-23

**Authors:** Daler Abdusamadzoda, Djamshed A. Abdushukurov, Octavian G. Duliu, Inga Zinicovscaia

**Affiliations:** 1Institute of Water Problem, Hydropower and Ecology of Academy of Science, 14a Ainy Str., Dushanbe 734042, Tajikistan; abdushukurov.dj@gmail.com; 2Frank Laboratory for Neutron Physics, Joint Institute for Nuclear Research, 6, Joilot Curie Str., 141980 Dubna, Russia; o.duliu@upcmail.ro (O.G.D.); zinikovskaia@mail.ru (I.Z.); 3Department of Structure of Matter, Earth and Atmospheric Physics and Astrophysics, Faculty of Physics, University of Bucharest, 405, Atomistilor Str., 077125 Magurele, Ilfov, Romania; 4Horia Hulubei R&D Institute for Physics and Nuclear Engineering, 30, Reactorului Str., 077125 Magurele, Ilfov, Romania

**Keywords:** Zarafshon Valley, exposed and unexposed soil and sediment, instrumental neutron activation analysis, pegmatite and granitoid, Djidjikrut, Yagnob, Fondarya, Tajikistan

## Abstract

This study discusses contamination of soils and sediments with trace elements such as Mn, Ba, W, V, Co, Cr, Zn, Ni, As, Sb Hg as well as Th and U, the influence of natural and anthropogenic factors on the distribution of elements and the ecological state of the Zarafshon Valley. The elemental composition of 116 soil and sediment samples were analyzed by the neutron activation analysis. The calculation of the geoaccumulation index (*Igeo*), contamination factor (*CF*) and pollution load index (*PLI*) showed that some places in the investigated region with developed the industrial zones (around mining and processing plants of “Anzob”, “Konchoch”, “Kumargi bolo” and “Mogiyon”) are mostly polluted by As, Sb, Hg, and in rare cases, the high concentration of W and V were determined. In addition, they were considered the distribution of radioactive elements–thorium and uranium and their ratio in the soil and sediments. Moreover, in the investigated area, strong anomalies of Th and U were not found. It turned out that the content of Th and U are local in nature and do not have a noticeable effect on the environment.

## 1. Introduction

The assessment of soil and sediment contamination by trace metals is a priority in the study of water bodies. The danger of increased content of toxic metals in the ecosystem is due to the fact that most metals have high biological activity, and their high concentrations can cause a toxic effect in vivo. In addition, toxic metals are slightly susceptible to biodegradation, and being in the biogeochemical cycle, they accumulate and are essentially not removed from the system. In this regard, constant monitoring is a primary and very important task to assess the toxic metal content in coastal soils and sediments of the rivers, especially in urbanized or industrial areas with a high level of technogenic load.

Coastal soils and sediments of rivers serve as a depositing medium of accumulating pollutants. Their chemical indicators reliably reflect the ecological condition of rivers and become of paramount importance in determining the degree of anthropogenic pollution of aquatic ecosystems, as well as their initial state, which are necessary for subsequent ecological and geochemical monitoring of the river network. However, in the sediments of rivers with a high flow rate, indicators of metal content do not reflect the complete picture of the pollution of the biota with toxic metals [1]. This is typical for the rivers of the Zarafshon Valley with a high declivity. As a rule, the concentration of toxic metals in water is lower than in sediments. This is largely determined by their rapid transition from a dissolved state into suspended matter, which has a high sorption capacity, and vice versa. Therefore, bottom sediments in rivers accumulate information about heavy metal pollution in the water [2,3].

Soil and sediment pollution with toxic metals (TMs) has been attracting increasing interest. However, assessment of eco-environmental and human risks, particularly in rapidly industrialized and urbanized areas of the Zarafshon Valley, remains limited. The Tajikistan Government is working on an international framework on “Water for Sustainable Development, 2018–2028”. One of the key factors of this project is to establish effective management of the prevention of water resource contamination. Heavy metal pollution is the main aim of the discussion concerning the quality of soil and sediments of rivers, as well as health and remediation in Tajikistan.

The Zarafshon is a transboundary river between Tajikistan and Uzbekistan, and its purity, especially water, is an area of interest for both countries. In the aquatic systems of rivers, the processes of sorption of toxic metals from water into bottom sediments and the reverse process of desorption—from bottom sediments into water are constant and occurs everywhere. The migration of toxic metals from bottom sediments into water depends on many parameters, both chemical and physical. The chemical composition of bottom sediments is actively affected by the geological and geochemical features of the area (geogenic), as well as anthropogenic impacts. During floods, spring waters actively wash away particles of rock and soil into the water, thereby forming bottom sediments of rivers. In addition, strong winds are capable of raising dust from the surface of the earth, which later settles on the surface of rivers.

It should be noted that bottom sediments in the valley are mainly formed due to the washout of soil rocks. Moreover, for mountain rivers, the formation of bottom sediments mainly occurs due to the washing away of rocks, which consists of quartz sand and feldspars. Feldspars do not have sufficient hardness. They can be easily broken down and are carried away downstream of rivers, forming loams in a more calm flow.

The Zarafshon River Basin consists of several zones with different geological composition. According to Vinogradov, the Zarafshon River Basin is divided into Turkestan-Alai, Turkestan-Zarafshon and Zarafshon-Gissar tectonic belts [4]. In the Zarafshon-Gissar Belt, there are widely developed deposits of Au, Ag, Sb, Hg, W, Sn, Pb, Zn and other types of minerals. The mercury-antimony geochemical belt covers the ore deposits of Shing-Magiyan, Konchoch, Djidjikrut, Yagnob-Zakhob [5]. Because of this, in the valley of Zarafshon mining industry is developed. Most of them are based on the extraction and processing of non-ferrous metals. Among them, the “Anzob” mining and metallurgical enterprises (AMME) have been operating for over 60 years. The “Taror” mining and metallurgical enterprises (TMME) have operated for 80 years in the last century. Recently, mining activity is underway at the Kumargi bolo and Konchoch gold deposits. AMME specializes in the production of antimony-mercury; TMME focuses on gold production. In Kumargi bolo and Konchoch, the complex mining of gold, antimony and mercury is carried out.

The activity of mining enterprises leads to soil and sediment pollution in investigated regions. In the Djidjikrut, Yagnob, Mastchohi kuhi regions, there are very small and low-fertility lands that are suitable for agricultural use. Most of the people use transported soil for farming. In many places, these lands are located above rivers and cannot be irrigated.

This study is the second part of a study of the distribution of chemical elements in the soil and sediment of the Zarafshon Valley. Detailed information on the geology and geochemical features of the region under study is described in the first part of this study, which is devoted to the sediment and soil geochemistry of the Zarafshon Valley, northwestern Tajikistan. The main goal of the study is to study the purity of the soil, the presence of toxic elements in the Zarafshon Valley under conditions of increased anthropogenic and natural impacts. As noted above, industrial activity is developed in this region, and water-treatment facilities and open wastewater storage of these industrial plants are located close to rivers. In this regard, there is always a high risk of pollution of rivers with toxic metals. At present, monitoring the metal pollution of transboundary rivers such as Zarafshon is an important task.

## 2. Materials and Methods

### 2.1. Sampling and Analysis

A total of 116 soil and sediment samples were collected in the investigated area using special equipment AMS 12 (multi-stage sediment sampler, produced by “AMS, Inc. American Falls” ID 83211, Waltham, MA, USA), equipped with tubular glass sampler [6]. Unconsolidated sediment, 10 to 20 cm-deep, were assembled directly from the riverbeds during an abundant-water (i.e., highwater) period. From each sampling site (Figure 1 and Table 1), we collected three subsamples in an area of 10–15 m^2^ and mixed them to make a composite sample. In order to avoid possible contamination and minimize the risks of adverse consequences during sampling, the top layers of soil and sediment were removed using a diluted acid-washed plastic spatula. In addition, each sampler was rinsed intensively with clean water and soaked with single-use cellulose napkins prior to reuse.

Consisting of thin grayish sand and previously cleaned from any organic or plant detritus, the soil and sediment samples were placed into the plastic container with cooled refrigerant and were kept at a 4 °C. In the laboratory, all samples were finally cleaned of alien materials and dried at 70 °C in a drying oven equipped with a venting system until constant weight. Then, the samples were sieved through a 0.425 mm (42 mesh) sieve, crushed, homogenized and sent to the Laboratory of Neutron Physics. Frank of the Joint Institute for Nuclear Research (JINR) to analyze using instrumental neutron activation analysis (INAA). In clean laboratory conditions, 10 g of each sample were weighed again and homogenized in a planetary ball mill PULVERISETTE 6 (Fritsch Laboratory Instruments GmbH, Germany) at 400 rpm for 15 min. For more reliable results and to increase the clarity and accuracy of measurement from each sample, six aliquots of about 0.3 g were selected to perform an independent analysis. Each aliquot was placed in a polyethylene bag for the determination of short-living isotopes (three aliquots) and in aluminum foil (three aliquots) for long-living isotopes. Irradiation of samples was performed by using the “REGATA” installation at the IBR-2 reactor of the JINR FLNP in Dubna. The gamma spectra of each sample were measured using a high purity germanium radiation detector (HPGe) detector with a resolution of 1.9 keV for the 60Co 1332 keV line. The measured gamma spectra were analyzed using the Canberra software Genie-2000. The gamma spectra of short-living isotopes were immediately recorded and processed after irradiation. In the case of the long-term irradiation, the gamma spectra were recorded and processed twice at 4 and 20 days after irradiation, respectively. This is the time required for the complete disintegration of short-lived radioisotopes. Using the appropriate software developed by the Frank Laboratory of Neutron Physics, the concentration of each element was calculated using the corresponding software developed by the Frank Laboratory of Neutron Physics [7,8]. This software enables calculating the content of each element and the associated combined standard uncertainty (CSU) [9], and also takes into account statistical errors, the influence of the measurement geometry, detector efficiency and uncertainty of each analyzed element in all certified standard materials (CSM) used for calibration. All activated samples after gamma spectroscopic measurements were placed in the lead shielded storage with low activity.

Quality control was provided by parallel analysis of the standard reference materials (SRM) of the National Institute for Standard and Technologies (NIST). The following standard materials were used to perform the quality control: for short-life isotopes—1633c (coal fly ash), 667 (estuarine sediment), 2710 Montana soil, 1547 peach leaves for long-life isotopes of sediment samples—2709 (trace elements in soil), 1632c (trace elements in coal), 690CC (calcareous soil); for long-life isotopes of soil samples—2709a (San Joaquin soil), 1632c (trace elements in coal), AGV2 (andesite). To create a group, standard samples (GSS) were used for all mentioned CRMs, which were irradiated along with the analyzed samples. More detailed information on the calculation procedure is described in [7,10]. Overall uncertainties in determining of mentioned elements were no greater than 15%.

### 2.2. Pollution Assessment

In the study of the assessment of the ecological state of the investigated area, special attention was given to the distribution of toxic elements (As, Sb, Hg, V, Cr, Mn, Co, Ni, Zn, Ba, W). In order to assess the level of contamination, we used the most frequently reported in literature indices, i.e., geoaccumulation index (*Igeo*) [11], contamination factor (*CF*) [12] and pollution load index (*PLI*) [13]. Since the studied samples were conditionally divided into two parts, the *Igeo*, *CF* and *PLI* were calculated for each group separately. The contamination factor (*CF*) was calculated as follows:(1)$$CF_{i}=\frac{C_{i}}{B_{i}}$$
where *C_i_* and *B_i_* is the geochemical background content of the same element in reference materials, in our case, the UCC for soils and SMWR for sediments, respectively. According to [12], a *CF* less than 1.0 shows no contamination, a *CF* between 1 and 3 indicates moderate contamination, between 3 and 6, considerable contamination, which for a *CF* greater than 6 is considered very high.

The *PLI* [13] was used to characterize the ecological condition of the investigated area. With respect to *CF*, the *PLI* provides more global information concerning the local contamination. Hence, the *PLI* for a number of n contaminants is defined as the geometric mean of the individual *CF*:(2)$$PLI=\sqrt[{n}]{\prod_{i-1}^{n}CF_{i}}$$
where *CF_i_* refers to the *i*-th element. By following the recommendations provided in [13], the environment could be interpreted as polluted if *PLI* ≥ 1 and as unpolluted if *PLI* < 1.

Another index that assesses the ecological condition is the geoaccumulation index (*Igeo*). As proposed by Muller [11], a common approach for estimating the enrichment of metal concentrations above background or baseline concentrations is to calculate the *Igeo*_,_ defined as:(3)$$Igeo=\log_{2}\left[\frac{C_{i}}{1.5B_{i}}\right]$$
where *C_i_* is the experimentally determined content of the element n in the investigated medium, and *B_i_* is the geochemical background content of the same element in reference materials, in our case, the UCC for soils and SMWR for sediments, respectively. A factor of 1.5 is used to compensate for possible variations, which may be attributed to lithological variations in the soil and sediment. If, on one hand, the *Igeo* shows the enrichment of soils by toxic metals, then, on the other hand, the values of *Igeo* for sediments are the quantitative measure of toxic metal pollution in the aquatic system. Basically, the *Igeo* is a single metal approach to quantify pollution of sediments and may be used when the concentration of toxic heavy metal is 1.5 or more times greater than their lithogenic background values [14].

According to Muller [11], the *Igeo* was distinguished into seven classes: *Igeo* ≤ 0, class 0, unpolluted; 0 < *Igeo* ≤ 1, class 1, unpolluted to moderately polluted; 1 < *Igeo* ≤ 2, class 2, moderately polluted; 2 < *Igeo* ≤ 3, class 3, moderately to strongly polluted; 3 < *Igeo* ≤ 4, class 4, strongly polluted; 4 < *Igeo* ≤ 5, class 5, strongly to extremely polluted; and *Igeo* > 5, class 6, extremely polluted.

## 3. Results

The content of 38 elements—major elements (rock-forming) Si, Ti, Al, Fe, Mn, Mg, Ca, K, Na and trace elements Sc, V, Cr, Co, Ni, Zn, As, Br, Rb, Sr, Zr, Sb, Cs, Ba, La, Ce, Nd, Sm, Eu, Gd, Tb, Tm, Yb, Hf, Ta, W, Hg, Th, U were determined. The results of the concentrations of all elements in 116 soil and sediment samples are shown in [15], along with reference values of the upper continental crust (UCC) [16] and the chemical composition of the suspended matter of the world’s rivers (SMWR) [17].

As this research is focused on the assessment of the ecological state of the investigated region, only the toxic elements such as V, Cr, Mn, Co, Ni, Zn, As, Sb, Ba, W, Hg, as well as Th and U (Table 2) are discussed. The content of these elements exceeded the levels established by national regulations [18] and neighboring Russian Federation [19], and they can be considered as contaminants.

Despite weak radioactivity, natural uranium and thorium are toxic, and their high concentration in soils increases the radioactive background. When ingested through the use of vegetation or by air–water, uranium and thorium act on all organs, being a general cellular poison. The interaction of uranium, like many other toxic metals, is practically irreversible. It may bind to proteins, primarily to the sulfide groups of amino acids, disrupting their function. The molecular mechanism of action of uranium is associated with its ability to suppress enzyme activity. Thorium is weakly toxic, but as a naturally radioactive element, it contributes to the natural background of an organism’s irradiation [20].

Moreover, for a better description and comparison in the column chart, the average values of soil were normalized by UCC and average values of sediment by SMWR (Figure 2). According to our data provided in [15] and (Table 2), the coefficient of variation of experimental data varied between 29% for Th and 660% for As in soil samples and between 44% for Th and 869% for Sb in sediment samples. As well as for Hg which contents were very high, the coefficient of variation was 369% in soil and 534% in sediments, respectively. The high values of standard deviations (Figure 2) and coefficients of variation (Table 2) for As, Sb and Hg are associated with their increased concentrations in some regions with anthropogenic impacts (for example, around AMME, Konchoch, Mogiyon and Kurmagi bolo).

## 4. Discussion

To appreciate the impact of natural and anthropogenic sources on the distribution of elements, all sampling zones were conditionally divided into two groups: the first ones includes samples unexposed to anthropogenic impacts (92 soil and 91 sediment samples, respectively), which were collected directly on the banks of the lakes, rivers and mountain tributaries, where any anthropogenic impact is absent (Figure 3A). It should be noticed that the mountain bedrocks located directly in the sampling sites (e.g., pegmatite and granitoids) have a great impact on the elemental content of soil and sediment. The second group includes samples that were exposed to anthropogenic activity (24 soil and 25 sediment samples, respectively). These samples were collected around road tunnels, near the industrial zones, coal mining, mines and adits that appeared during the geological exploration. Some of the adits related to the mining and processing plants are still under exploration. After dividing into two groups, all data were separately normalized using reference values. As expected, the impact of anthropogenic sources was still quite high. In the unexposed soil and sediment samples (Figure 3A), the content of As, Sb, Hg and partially Zn, Ba and U remained still higher than reference values.

It is known that most metals in a mobile form (cations or oxyanions) exhibit high toxicity and mobility (migration ability). In this research, a study to establish the concentration of mobile forms of toxic metals was not carried out due to the limitation of the laboratory facilities. This approach would make it possible to more objectively assess the ecological condition of soil and sediment compared by the study of the mass fraction of all forms.

The high content (except As, Sb, Hg) of V, Cr, Ni, Zn, Cs and W, as shown in (Figure 3B) indicates soil exposure to anthropogenic impacts. The high content of V was determined in soil samples collected around the Anzob mining and processing plant (i-06, i-15, i-16 and i-17), in the gold-mining combine of Kumargi bolo (i-39 and i-40), around geological prospecting objects (adits) of Chore (i-42, i-44, i-45), and especially in the soil samples collected around the Shahriston Tunnel. During the tunnel construction, the shredded solid rocks were taken out and scattered on the surface over more than 7500 m2 near the Shahriston tributary. Thus, this may be one of the main sources of the high content of elements in soil and sediment in Shahriston. The content of V in soil of these areas (i-77, i-78 and i-79) varied between 154 and 14,500 mg/kg [15].

In the case of sediments exposed to anthropogenic impact, except As, Sb, Hg, high content of Ni, Zn, W and U was also determined. Though the average values of V and Cr were close to the reference values (Figure 3B) and their content varied between 27.3 and 655.0 and 20.5–147.0 mg/kg, respectively [15]. As can be seen in (Figure 3B), anthropogenic factors have an influence on the content of some toxic elements. However, even in the absence of any anthropogenic loading, the content of Sb, As, and Hg remains high. This is especially evident in the zones of mercury–antimony deposits. Except for this, the influence of geochemical features leads to increasing the content of natural radioactive elements–thorium (Th) and uranium (U) in all studied samples. Their average values were close to reference values (Figure 3A,B), but if we consider each sample individually, it can be clearly seen that in some samples, the content of Th and U is higher than reference values [15].

The average values of elements in unexposed soils and sediments, illustrated in (Figure 4A—red line) indicates that there was a good linear correlation between them, which can be better described by a Pearson’s correlation coefficient equal to 0.99 at *p* < 0.01. In the absence of normal distribution, the nonparametric Spearman correlation coefficient is the optimal approach for better interpretation of data. The relative similarity in terms of the content of investigated elements can evidence the practical identity of bottom sediments and soils. However, in the case of anthropogenically exposed soil and sediment (Figure 4A—blue line), the correlation coefficient equal to 0.62. It can be seen that the statistical relationship between the average values of exposed and unexposed soil and sediment are very different. This indicates great changes in the elemental balance in the anthropogenically exposed samples.

In addition, the correlation between the anthropogenically exposed and unexposed samples (Figure 3B lines red and blue) confirms that the higher contents of trace metals are in anthropogenically exposed samples. As it can be seen in (Figure 4A), the values of Hg, Sb and V of anthropogenically exposed samples do not lie on the correlation line. It can be explained by the large variation in their values (Table 2). In addition, the correlation between the anthropogenically exposed and unexposed samples (Figure 3B lines red and blue) confirms that the higher contents of trace metals are observed in anthropogenically exposed samples.

One of the important indicators of soil pollution is the ratio of thorium to uranium (Th/U). The Th/U ratio of 3–5 is observed in the overwhelming majority of soils in different regions, countries and continents, regardless of their types of origin [20,21]. The result of our study showed that the Th/U ratio in soils and in bottom sediments is predominantly at the regional level of 3.5–5 (Figure 5). In rare cases, the Th/U ratio in the studied soils varies from 1.3 (at Th/U = 11.6/8.6 mg/kg) to 6.2 (at Th/U = 9.3/1.5 mg/kg) and in bottom sediments from 1.1 (at Th/U = 3.3/3.1 mg/kg) to 6.2 (at Th/U = 9.8/1.6 mg/kg), respectively.

In soils in the zone of anthropogenic impact (around the up and downstream of the Djidjikrut River, in the Konchoch adits and Mogiyon), the Th/U ratio is less than 3. Low Th/U ratios are observed around the upstream of Sarvoda, Mastchohi kuhi, Haftkul and the downstream of the Zarafshon River (in the Panjekent city), where practically there is no anthropogenic impact. In addition, low values of the Th/U ratio are observed in the main part of samples of bottom sediments. However, in the samples of bottom sediments from the lower reaches of the Shing and the Konchoch adit, the Th/U ratio is 6.0 (at 9.8/1.6 mg/kg) and 6.2 (at 6.6/1.1 mg/kg), respectively. According to [20,21], if the Th/U ratio is less than 2, and even less than 1, then it can be asserted that the studied samples do not belong to magmatic formations but to metasomatic or metasomatically transformed rocks. In addition, the presence of igneous rocks (Th/U >5) is also observed in some places.

The higher content of thorium and uranium content over UCC in soils is observed around the lower reaches of Djidjikrut and Yagnob rivers, around tributary Pete, in the Mastochohi kuhi region, Shakhriston and Mogiyon. In the sediments, a higher thorium and uranium content was found in the lower reaches of the Djidjikrut, in the tributaries of the Shakhriston and Mogiyon. The exceedance in all the indicated places varies 1–2.5 times, which is typical for these regions. The serious geochemical changes and health hazards by uranium and thorium are not observed in soils and sediments.

According to (Figure 6A and Appendix A) the *CF* values (more precisely comparing by UCC) in the unexposed soils are moderately contaminated by V, Cr, Mn, Co, Ni and Zn, moderate to considerable contaminated by Ba and W, and considerable to very high contaminated by As, Sb and Hg. However, this is exclusively depending on the geochemical features of these regions. In the case of anthropogenically exposed soils (Figure 6C,D and Appendix A), the *CF* varies between 0.76 and 876 for Hg, 1.9–1237 for As, 4.05–6400 for Sb and 0.58–149 for V. The minimum *CF* values for these elements (Figure 6A and Appendix A) were found in the upper reaches of the Zarafshon (Mastchohi kuhi) Yagnob, Sarvoda rivers (including the Alovdin lakes), as well as in upstream of Mogiyon river, which are assessed as clean zones.

In (Figure 6B and Appendix A) shows the values of *CF* for unexposed sediments. There is no contamination in unexposed sediments with Mn and Co. In addition, according to average values of *CF*, there is no contamination with V, Cr, Ni and Zn; however, in individual places, values of *CF* indicate moderate contamination. Average and maximum values of *CF* for As and Sb indicate moderate to very high contamination, for Ba moderate to considerable contamination and for W moderate contamination. For Hg, even the lowest values of *CF* indicate moderate contamination.

For anthropogenically exposed soils (Figure 6C and Appendix A), the average values of *CF* indicate very high contamination with V and moderate contamination with Cr, Ni, Zn, W, but for Mn, Co and Ba, the average values of *CF* lie under the unit which indicates on the absence of contamination. The maximum values of *CF* for V and W relate to the soils collected around Shahriston tunnel and Konchoch mine adit, respectively and indicate very high contamination. In addition, the high values of Cr, Ni and Zn for Konchoch mine adit point out moderate contamination. In the case of As, Sb, and Hg (Figure 6D), the values of *CF* vary in the following range 1.9–1237, 4–6400 and 0.7–876, respectively. The minimum values of *CF* for Hg are observed in Saritag and Chore-3 (adit) 0.7–0.9, respectively. The average and maximum values of *CF* in all anthropogenically exposed soil indicate very high contamination.

The *CF* values of V, Cr, Ni, Zn and Ba indicate an absence of contamination in anthropogenically exposed sediments (Figure 6E and Appendix A). However, the average and maximum values of *CF* for W are higher than in anthropogenically exposed soil, which indicates considerable and very high contamination. In addition, anthropogenically exposed sediments (Figure 6F and Appendix A) show characteristically very high contamination with As, Sb and Hg, which *CF* values vary in the range 0.14–240, 0.49–3935 and 2.2–1818, respectively. In spite of the anthropogenic impacts in sediments from upstream of Djidjikrut River (around Anzob mining company), there is no contamination with As (CFAs = 0.14), and *CF* values for Sb and Hg indicate very high contamination. The minimum values of *CF* for As and Sb (CFAs = 0.8 and CFSb = 0.4) were found in Saritag and indicated an absence of contamination. The average and maximum values of *CF* for As, Sb and Hg in all anthropogenically exposed sediments indicate very high contamination.

The values of *PLI* for soils and sediments in the investigated region are shown in (Appendix A). In the case of unexposed soil and sediment samples (Figure 6A,B), the *PLI* values lower than unity were observed only upstream of Yagnob, Iskandardarya, Alovdin lakes and Sarvoda, as well as at upstream of Zarafshon, Shing and Mogiyon rivers, which can be considered unpolluted. All anthropogenically exposed samples may be considered as polluted since the *PLI* ≥1.

The minimum, average and maximum values of *Igeo* for exposed and unexposed soil and sediment samples are depicted in (Figure 7, Appendix A). How it is seen from (Figure 7A and Appendix A) the minimum, and average values of *Igeo* in the unexposed soils indicate an absence of any contamination with V, Cr, Mn, Co, Ni, Zn, Ba, W, but their maximum values (except Mn) are higher than zero, indicating uncontaminated to moderately contamination (class II) and moderately contamination (class III) environment. The average and maximum values indicate non-contamination to moderate contamination (class II) and strongly to extremely contamination (class V). In the case of Sb and Hg, their average and maximum values indicate moderately to strongly contaminated (class III) and extremely contaminated (class VI).

In the case of anthropogenically exposed soils (Figure 7B and Appendix A), more intensive accumulation is observed for As, Sb and Hg, which average values indicate moderately to strongly contamination for Hg (class III), strongly contaminated for As and from strongly to extremely contamination (class V) for Sb. In addition, the maximum values of *Igeo* for V, As, Sb and Hg reach 6.6, 9.6, 12 and 9.1 and indicate extreme contamination (class VI). The maximum values of *Igeo* for Cr, Co, Ni and Zn indicate uncontaminated to moderately contaminated (class I) and for W strongly contaminated (class III).

It should be noted that there is no mining and industrial activity in the Mastchohi kuhi, and this region can be considered as clean. However, in some soils and sediments (Figure 7A,C and Appendix A) collected in this region, the *Igeo* for Hg reaches values higher than 3. In the case of sediment from tributary Obi Sara in Panjakent *Igeo* for Sb and Hg varies between 1.42 and 2.0 and 1.42–2.74, respectively. In the Mogiyon River, the *Igeo* for As and Sb only in one sapling site reached 3.75 and 5.79, respectively. Downstream of the Mogiyon River, the *Igeo* for Hg varies between 2.0 and 3.74, and downstream of Zarafshon River for Sb and Hg, it varies between 2.06 and 2.57 and 2.0–3.22, respectively.

It is important to note that the *Igeo* values for toxic elements in sediments depend on their chemical forms, pH of water in the investigated system, solubility equilibrium of these metal compounds in water and many other chemical and biochemical factors. Except it, there are many physical factors—river slope or tributaries, volume and speed of water flow, narrowing and expansion of the riverbed and others, which can affect the accumulation of toxic elements in sediments.

High concentrations of As, Sb and Hg determined in the studied region could be explained by the irrigation of local fields with mine water from the adit. For example, the pH of water in the Konchoch mine adit is 2.7, which is equivalent to table vinegar. Such water dissolves many minerals and brings out to the surface. At the downstream of the adit located a different sizes plot of land on which local residents grow wheat and grass for mowing. In addition, willows and poplar, as well as fruit trees, grow along a polluted irrigation ditch. Furthermore, it can greatly affect the elemental balance and the water quality of Iskandarkul, which is located 3 km from these places. In this case, the observed discrepancy between the level of contamination assessed by *Igeo* and *CF* could be explained by the presence of some unitary criteria to characterize the contamination level.

To evidence the possible sources of anthropogenic or natural contamination, the corresponding matrix of the Spearman’s correlation coefficients were calculated (Table 3 and Table 4 for soils and sediments, respectively). The data reproduced in the correlation matrix (Table 3A) confirms that the unexposed soil samples were noticed a good correlation only between Mn-Ba (r = 0.70). However, in soils exposed to anthropogenic impacts there are significant correlation between Cr-Ni (r = 0.72), Cr-Zn (r = 0.70), Cr-As (r = 0.79), Cr-Sb (r = 0.74), Cr-W (0.73) as well as between Mn-Co (r = 0.80), As-Sb (r = 0.96), As-W (r = 0.95) and Sb-W (r = 0.89) (Table 4B) [22]. Despite the high concentrations of Hg in soil samples, it does not correlate with any element (Table 4A,B) even with Sb and As. In our opinion, it is due to the relatively low content of Hg in rocks. According to [5,23], the ratio between Sb and Hg in the Djidjikrut ore belt is 100 to 1, respectively.

In the case of unexposed sediments (Table 4A) there are significantly correlations between Cr-Ni (r = 0.76), Mn-Co (r = 0.70), Mn-W (r = 0.70) and also Sb-Hg (r = 0.97). In addition, in anthropogenically exposed sediments it is noticed a significantly correlation between Cr-Zn (r = 0.72), Mn-Co (r = 0.77), Mn-Ni (r = 0.70), Mn-W (r = 0.76), Co-Ni (r = 0.75) as well as Sb-Hg (r = 0.97). A significantly positive correlation suggests also that these metals are redistributed in the soil and sediment by the same processes or had a similar source.

## 5. Conclusions

In order to assess the ecological state of the Zarafshon Valley, the elemental content of coastal soils and sediments of rivers as well as small and large tributaries in this region was investigated. The Zarafshon Valley is exposed to strong anthropogenic impacts from mining enterprises, which actively influence the ecological state of the region. In the course of the study, the concentrations of 13 elements, first and second hazard classes, As, Ba, Co, Cr, Mn, Ni, Sb, Hg, V W, Zn and also Th and U, were determined by NAA. Mainly in terms, the anomaly zones were observed by contents of As, Sb, Hg and partly V, W and Zn. It should be noted that the geochemical features of this region cannot be ruled out [10,15]. At the same time, the upper reaches of the Fann Mountains and Mastchohi kuhi can be considered as clean zones. The most heavily polluted is the lower reaches of the Djidjikrut river, below the Anzob mining and processing plant. Despite the excessively high concentration of toxic metals in the Djidjikrut River, their concentrations in bottom sediments after the AGOK (Yagnob and Fondarya) do not differ much from the overlying points along the Yagnob River. This can be explained by the fact that during floods and especially mudflows, river bottom sediments are washed away downstream and accumulate in the lower reaches of rivers.

The gross contents of Th and U showed almost the same variability of their contents both in soils and in sediment samples. The strong anomalies in the content of thorium and uranium were not observed in the studied regions. However, in isolated cases, the content of thorium and uranium exceed the upper continental crust values 1–2 times, indicating the peculiarities of rocks. The Th/U ratio mainly varies from 3 to 5. A decrease or increase of this ratio is typical for soils of industrial sites, their margins and in areas subjected to erosion.

Due to the fact that the sampling of soils and sediments were carried out at depths of 10–20 cm, at the moment, one can only assume that the technogenic genesis of investigated regions is within increased values. Nevertheless, areas with a clear excess of values require detailed studies, and it is the immediate task of further research. Thence, the considered data are insufficient to correct assessment of the ecological condition of the investigated region. To accomplish this task, it is necessary to carry out sampling of soils and sediments at different depths. Especially high contents of As, Sb Hg require careful examination using various physical-chemical or nuclear-physical methods. For a more accurate assessment of the pollution level and taking into account the mountainous of the investigated region, it will be better to use the values of the local background.

In addition, one of the important tasks in the future of this study is to determine the concentration of mobile forms—cations and oxyanions of toxic metals in soil and water samples, to establish the content of organic matter and the ratio of humic acids, the buffering capacity of soil–water system, acid–base characteristics of soils and sorption capacity of the soil-absorbed complex (SAC). Since from the point of view of ecogeochemistry, the main function of the SAC is to participate in complexation and exchange reactions with a toxic element. It makes it possible to determine the exchangeable forms of Ca, Mg, K, Na, as well as the hydrolytic acidity and pH of soils in different layers of the soil cover.

## Figures and Tables

**Figure 1 toxics-08-00113-f001:**
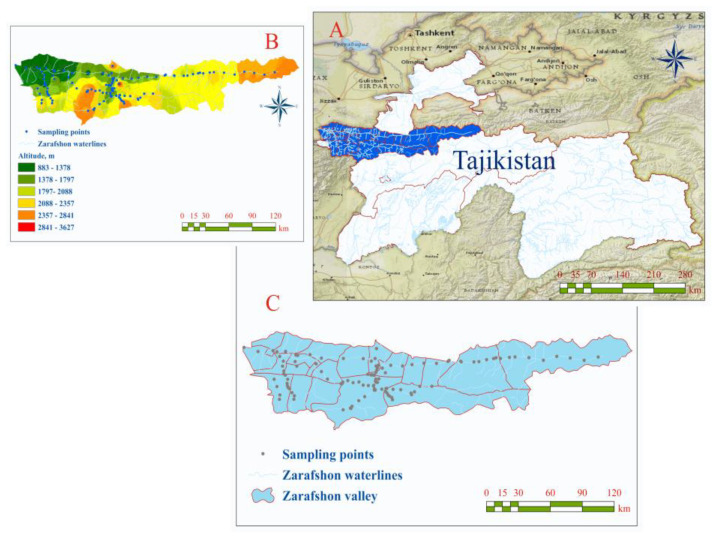
Investigated area location in the Tajikistan map (**A**); altitude (meter) of the investigated area, respectively by sampling points (**B**); soil and sediment sampling sites of Zarafshon Valley (**C**).

**Figure 2 toxics-08-00113-f002:**
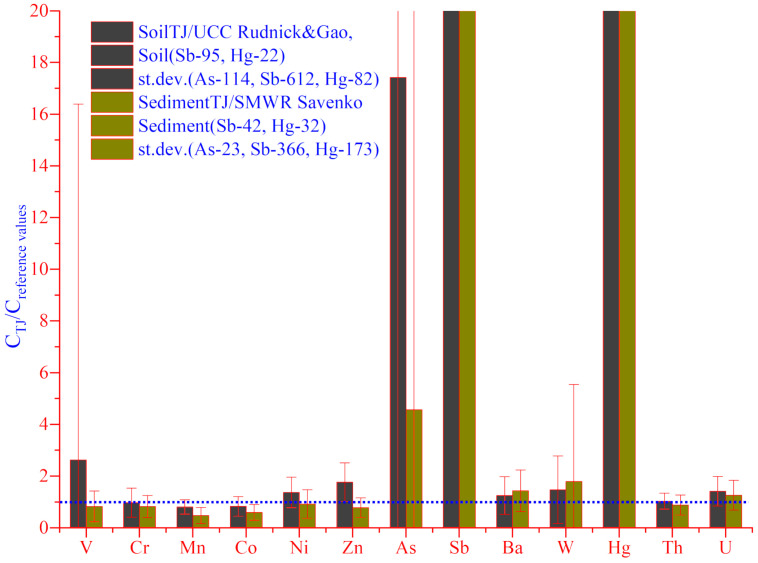
Column chart illustrating the distribution of investigated elements in soils and sediments (Mean ± St.Dev), normalized by reference materials.

**Figure 3 toxics-08-00113-f003:**
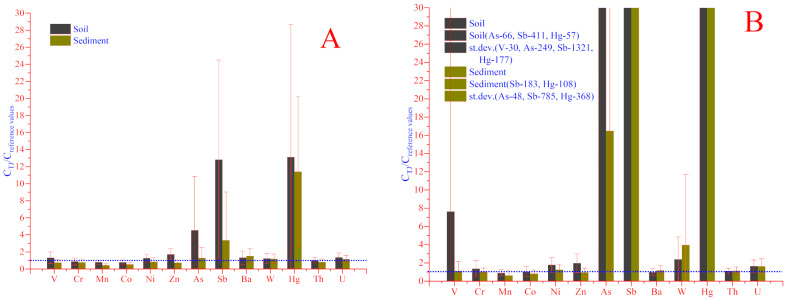
Normalized data of unexposed soil and sediment (**A**) samples, as well as soil and sediment exposed by anthropogenic impacts (**B**).

**Figure 4 toxics-08-00113-f004:**
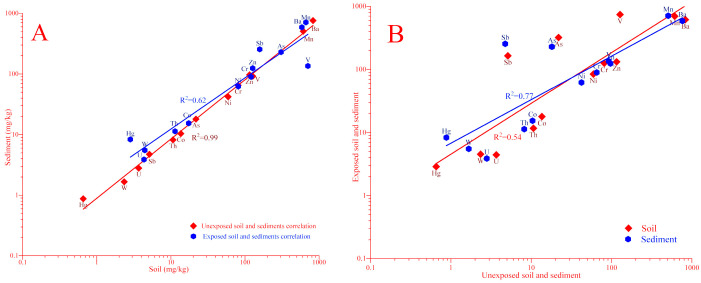
Correlation of average values of elements in unexposed soil and sediment (red line) and anthropogenically exposed soil and sediment samples (blue line) (**A**); as well as correlation between unexposed and anthropogenically exposed soil (red line) and unexposed and anthropogenically exposed sediment samples (blue line) (**B**).

**Figure 5 toxics-08-00113-f005:**
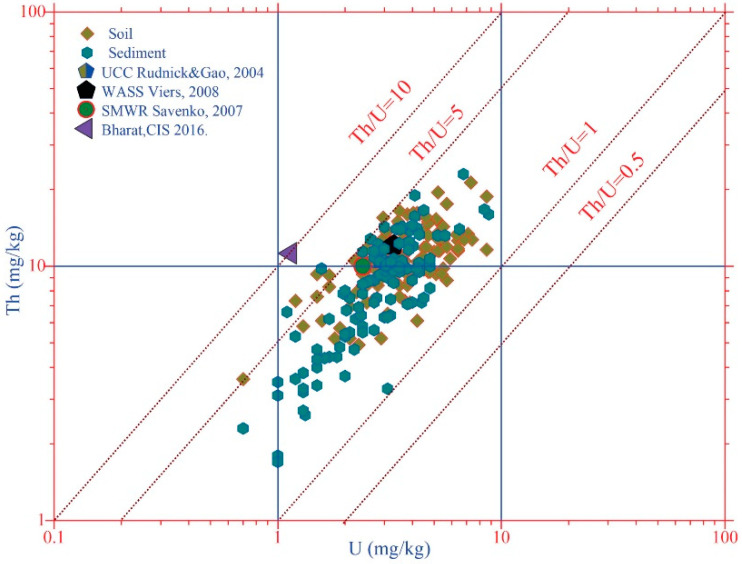
The distribution of the contents of thorium and uranium relative to their ratio in soils and sediments.

**Figure 6 toxics-08-00113-f006:**
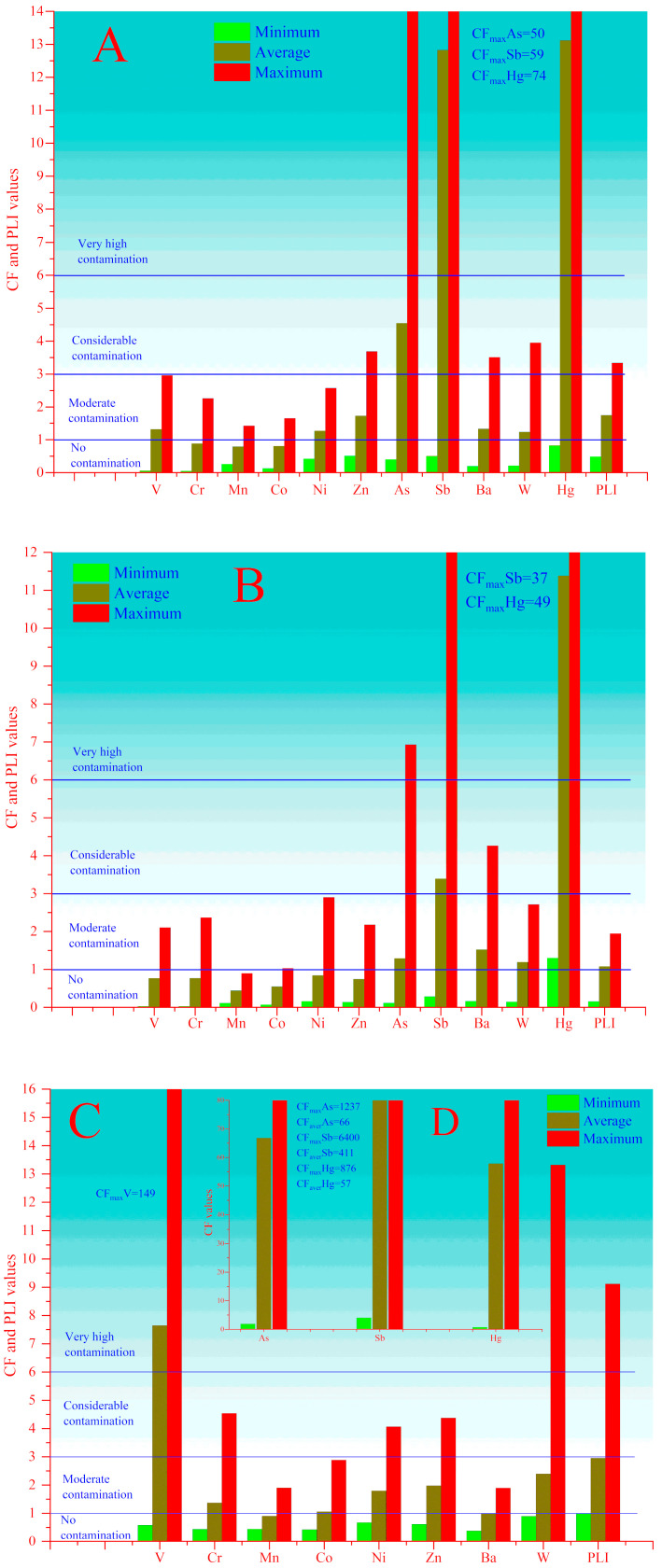
Values of *CF* and *PLI* of unexposed soil (**A**) and sediments (**B**), as well as soils (**C**,**D**) and sediments (**E**,**F**) related to anthropogenic impact.

**Figure 7 toxics-08-00113-f007:**
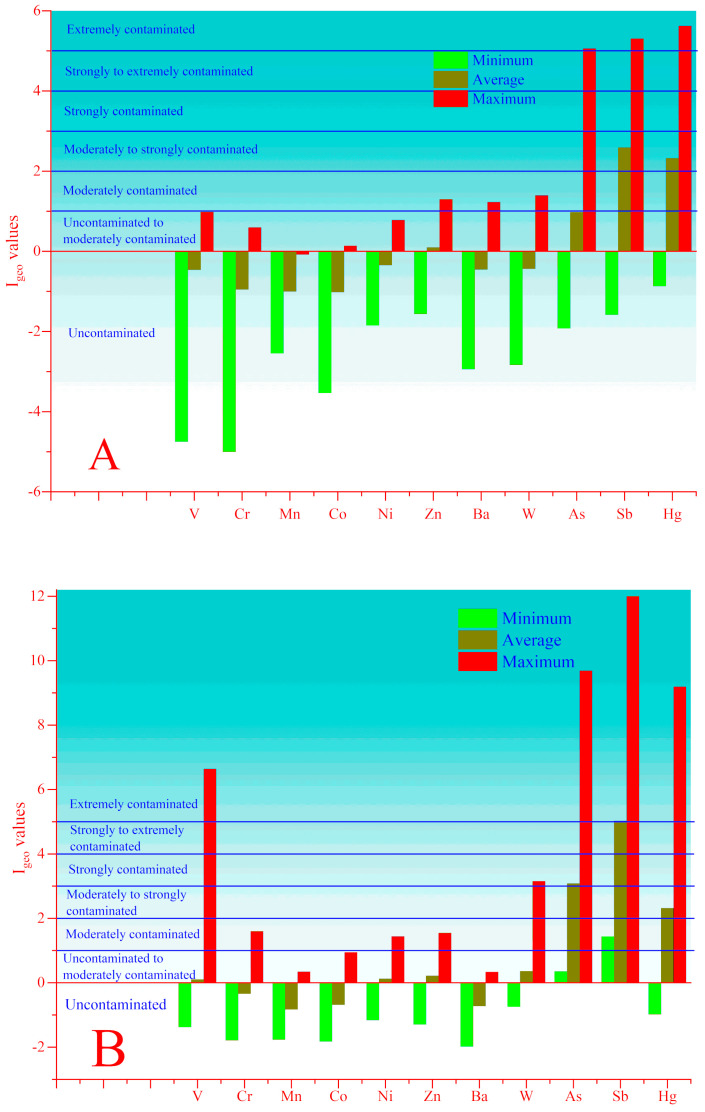
Values of *Igeo* for unexposed (**A**) and exposed to anthropogenic impact soils (**B**); as well as unexposed (**C**) and exposed to anthropogenic impact sediments (**D**).

**Table 1 toxics-08-00113-t001:** Geographical coordinates of sampling points.

Sampling Point Number	Longitude	Latitude	Altitude (m)	Sampling Point Number	Longitude	Latitude	Altitude (m)	Sampling Point Number	Longitude	Latitude	Altitude (m)
01	39.1091	68.6884	2726	40	39.2754	68.5476	1690	79	39.4406	68.5414	1682
02	39.1266	68.6829	2574	41	39.3136	68.4797	2148	80	39.4195	68.5093	1366
03	39.1379	68.6636	2444	42	39.3089	68.5062	1863	81	39.4392	68.4416	1306
04	39.1468	68.6518	2398	43	39.3086	68.5141	1753	82	39.4392	68.4416	1284
05	39.1645	68.6419	2318	44	39.3082	68.5287	1577	83	39.4408	68.2733	1226
06	39.1763	68.6274	2252	45	39.3061	68.5329	1580	84	39.4435	68.0878	1206
07	39.1995	68.6333	1742	46	39.3055	68.5339	1580	85	39.2749	68.1392	2142
08	39.2173	69.0211	2351	47	39.3448	68.5543	1466	86	39.3353	68.0878	1573
09	39.2168	68.9193	2215	48	39.3259	68.6284	3629	87	39.3978	68.0433	1267
10	39.1728	68.8744	2151	49	39.3632	68.6027	1782	88	39.4717	67.9918	1113
11	39.1621	68.8452	2090	50	39.3640	68.5611	1434	89	39.4665	67.8871	1072
12	39.1503	68.8500	2144	51	39.3291	68.5484	1638	90	39.3793	67.8800	1775
13	39.1897	68.7587	2070	52	39.3805	68.5496	1389	91	39.4159	67.8656	1315
14	39.1930	68.6935	1888	53	39.4527	70.4262	2732	92	39.4684	67.8728	1079
15	39.2000	68.6401	1739	54	39.4262	70.3076	2612	93	39.4973	67.7851	1035
16	39.1943	68.5983	1706	55	39.4445	70.1898	2517	94	39.1133	67.8553	2393
17	39.1875	68.5434	1657	56	39.4576	70.0762	2423	95	39.1410	67.8595	2106
18	39.0285	68.2713	2477	57	39.4264	69.9634	2330	96	39.1661	67.8390	1888
19	39.0564	68.3464	2194	58	39.4476	69.8412	2272	97	39.1973	67.8193	1811
20	39.0422	68.3328	2361	59	39.4532	69.7239	2267	98	39.2036	67.8107	1718
21	39.0395	68.3379	2457	60	39.4492	69.6763	2242	99	39.2101	67.8050	1701
22	39.1014	68.4036	2015	61	39.4404	69.6013	2158	100	39.2203	67.8008	1610
23	39.1340	68.4535	1820	62	39.4469	69.5635	2123	101	39.2408	67.7980	1565
24	39.1866	68.5349	1658	63	39.4399	69.5424	2075	102	39.2737	67.8028	1382
25	39.1964	68.5360	1644	64	39.4312	69.4836	1999	103	39.3069	67.7838	1282
26	39.2376	68.2543	2780	65	39.4158	69.4187	1985	104	39.1126	67.7336	2383
27	39.2375	68.2611	2780	66	39.4133	69.3768	1927	105	39.1478	67.7370	2144
28	39.2502	68.2689	2620	67	39.4138	69.2692	1919	106	39.2152	67.7119	1722
29	39.2629	68.3109	2488	68	39.4182	69.2649	1913	107	39.2737	67.6886	1474
30	39.2463	68.3558	2358	69	39.4256	69.1673	1875	108	39.3156	67.7702	1250
31	39.2521	68.4051	2166	70	39.4171	69.1568	1820	109	39.3396	67.7703	1222
32	39.2266	68.4410	1958	71	39.4012	69.0605	1742	110	39.3788	67.7595	1180
33	39.2232	68.4846	1808	72	39.4028	68.9540	1660	111	39.4194	67.7661	1099
34	39.2199	68.5276	1660	73	39.3915	68.8533	1619	112	39.4566	67.7359	1041
35	39.2252	68.5303	1626	74	39.3806	68.7734	1508	113	39.4951	67.7119	994
36	39.2486	68.5618	1895	75	39.3859	68.6739	1462	114	39.5029	67.6789	981
37	39.2494	68.5499	1710	76	39.3873	68.5703	1405	115	39.4967	67.5522	934
38	39.2588	68.5440	1589	77	39.5208	68.5534	2728	116	39.5295	67.4339	883
39	39.2769	68.5731	2029	78	39.5200	68.5549	2744				

**Table 2 toxics-08-00113-t002:** Average, maximum, minimum, median, standard deviation and coefficient of variation, regarding the distribution of 13 toxic elements in soils and sediments. For comparison, the corresponding values of the upper continental crust (UCC) and the suspended materials of world rivers (SMWR) are also provided.

	Average	Maximum	Minimum	Median	St. Dev.	Coefficient of Variation, %	UCC Rudnick and Gao, 2004
**Soils**							
V	254	14,500	5.4	125	1330	525	97
Cr	90.1	417	4.3	86.5	51.5	57	92
Mn	629	1470	199	622	211	34	774
Co	14.5	49	2.2	13.8	6.4	45	17.3
Ni	64.5	191	19.6	59	27.7	43	47
Zn	118	293	34	110	49.3	42	67
As	83.7	5940	1.9	18.4	548	658	4.8
Sb	38.1	2560	0.2	4.3	244	643	0.4
Ba	784	2190	122	622	452	58	624
W	2.8	25.3	0.4	2.3	2.5	89	1.9
Hg	1.1	43.8	0.01	0.4	4.1	369	0.05
Th	10.9	21.3	3.6	10.7	3.2	29	10.5
U	3.8	8.6	0.7	3.6	1.5	40	2.7
**Sediments**							**SMWR Savenko, 2007**
V	100	655	3.0	94.5	71.2	72	120
Cr	70.4	201	2.0	72.5	36.3	52	85
Mn	551	3650	129	528	361	66	1150
Co	11.5	45	1.3	11.2	5.8	51	19
Ni	46.4	154	7.8	38.3	27.4	59	50
Zn	101	283	18	94	48.8	48	130
As	63.9	3360	1.6	16.3	321	505	14
Sb	59	5510	0.4	2.5	510	869	1.4
Ba	720	2130	81	669	399	56	500
W	2.5	51	0.2	1.8	5.2	208	1.4
Hg	2.5	140	0.1	0.8	13.3	534	0.08
Th	8.9	23	1.7	9.3	3.9	44	10
U	3.0	8.8	0.7	3.0	1.4	46	2.4

**Table 3 toxics-08-00113-t003:** Matrix of the Spearman’s correlation coefficient of presumed contaminant elements in soil samples (in bold correlation coefficients at *p* < 0.05).

**(A) Unexposed Soils**											
	**V**	**Cr**	**Mn**	**Co**	**Ni**	**Zn**	**As**	**Sb**	**Ba**	**W**	**Hg**
V	1.00										
Cr	0.63	1.00									
Mn	0.52	0.47	1.00								
Co	0.58	0.59	0.57	1.00							
Ni	0.24	0.56	0.18	0.33	1.00						
Zn	0.59	0.51	0.50	0.50	0.27	1.00					
As	0.04	0.20	0.17	0.16	0.23	0.03	1.00				
Sb	0.05	0.33	−0.07	−0.01	0.35	0.00	0.31	1.00			
Ba	0.57	0.42	0.70	0.46	0.05	0.63	−0.13	−0.14	1.00		
W	0.28	0.28	0.22	0.35	0.14	0.17	0.46	0.53	0.16	1.00	
Hg	0.21	0.08	0.10	0.26	0.29	0.09	0.13	0.05	−0.01	0.14	1.00
**(B) Anthropogenic Exposed Soils**											
	**V**	**Cr**	**Mn**	**Co**	**Ni**	**Zn**	**As**	**Sb**	**Ba**	**W**	**Hg**
V	1.00										
Cr	−0.09	1.00									
Mn	0.03	−0.10	1.00								
Co	0.04	0.00	0.80	1.00							
Ni	−0.10	0.72	0.15	0.34	1.00						
Zn	−0.03	0.70	0.50	0.47	0.58	1.00					
As	−0.06	0.79	−0.23	−0.23	0.59	0.40	1.00				
Sb	−0.07	0.74	−0.27	−0.30	0.60	0.37	0.96	1.00			
Ba	0.40	0.25	0.41	0.50	0.24	0.59	−0.02	−0.03	1.00		
W	−0.08	0.73	−0.20	−0.13	0.58	0.39	0.95	0.89	0.01	1.00	
Hg	−0.07	−0.02	−0.11	−0.18	0.08	0.00	−0.04	0.21	0.08	−0.10	1.00

**Table 4 toxics-08-00113-t004:** Matrix of the Spearman’s correlation coefficient of presumed contaminant elements in sediment samples (in bold correlation coefficients at *p* < 0.05).

**(A) Unexposed Sediments**											
	**V**	**Cr**	**Mn**	**Co**	**Ni**	**Zn**	**As**	**Sb**	**Ba**	**W**	**Hg**
V	1.00										
Cr	0.41	1.00									
Mn	0.25	0.45	1.00								
Co	0.39	0.67	0.70	1.00							
Ni	0.26	0.76	0.54	0.65	1.00						
Zn	0.39	0.61	0.62	0.64	0.61	1.00					
As	−0.02	0.23	0.13	0.07	0.09	0.31	1.00				
Sb	−0.01	0.08	−0.06	0.11	0.21	0.14	0.07	1.00			
Ba	0.31	0.23	0.35	0.27	0.10	0.60	−0.13	0.01	1.00		
W	0.09	0.32	0.70	0.52	0.39	0.46	0.61	0.10	−0.02	1.00	
Hg	0.01	0.08	−0.07	0.17	0.23	0.16	0.04	0.97	0.02	0.09	1.00
**(B) Anthropogenic Exposed Sediment**											
	**V**	**Cr**	**Mn**	**Co**	**Ni**	**Zn**	**As**	**Sb**	**Ba**	**W**	**Hg**
V	1.00										
Cr	0.03	1.00									
Mn	−0.02	0.48	1.00								
Co	0.06	0.56	0.77	1.00							
Ni	−0.02	0.68	0.70	0.75	1.00						
Zn	−0.05	0.72	0.56	0.59	0.69	1.00					
As	−0.14	0.34	0.08	−0.07	−0.01	0.43	1.00				
Sb	−0.08	0.03	−0.13	0.06	0.30	0.16	0.02	1.00			
Ba	−0.01	0.28	0.15	0.38	0.38	0.59	−0.29	0.15	1.00		
W	−0.04	0.44	0.76	0.61	0.51	0.60	0.58	0.04	−0.05	1.00	
Hg	−0.08	0.04	−0.15	0.13	0.33	0.16	−0.03	0.97	0.19	0.02	1.00

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
