# Peer review of "Assessment of the Toxic Metals Pollution of Soil and Sediment in Zarafshon Valley, Northwest Tajikistan (Part II)"

_toxics, 2020, doi:10.3390/toxics8040113_

Round 1

Reviewer 1 Report

Review of “Assessment of the Heavy Metals Pollution of Soil and Sediment in Zarafshon Valley” by Abdusamadzoda et al submitted to Toxics

The study by Abdusamadzoda et al., investigates trace metal and metalloid (not heavy metals) in soils and sediments throughout the Zarafshon valley boradering Tajikistan and Uzbekistan. There is a lot of great data here but there are issues that need to be resolved before publication. One of the biggest issues is that the discussion section currently is an extension of the results section and do not interpret why the metal concentrations are enriched, what pathways caused the pollution, what soil or sediment properties allowed for enrichment, and exposure to plants, fish, birds, and humans are not adequately discussed. The authors briefly mention the concentrations are elevated but do not elaborate on what that means for ecosystems or humans. There are some nice figures, but the map needs to be redone to be readable, and the authors should remove the linear regression tables for more useful information like concentrations in polluted and unpolluted soils and potential use spatial mapping of the soil and sediment Igeo values with respect to mines and human residences.

The authors should compare the differences of elements in polluted and unpolluted soils, which is described in the discussion section but not shown in any figures. That would be more useful than the existing figure 7 and the correlation tables.

Metal and metalloid concentration data should be shown. It would be nice to have a figure or table in the manuscript that shows V, Cr, Mn, Co, Ni, Zn, As, Sb, Ba, W, Hg concentrations in soils and sediments separately, not just the Igeo values.

Further, the authors should consider if metals that exist as cations (Co, Ni, Zn, Ba) or oxyanions (Cr, As, V, U) are present in soils and sediments as anions can have greater mobility as soil materials (organic matter, clays, oxides) typically are negatively charged.

Be sure to use element abbreviations consistently throughout the manuscript.

Arsenic is a metalloid, not a heavy metal or trace metal.

Specific comments

Title: I recommend stating Tajikistan in the title. It will help global scientists find it.

Abstract

Line 16: Substitute ‘toxic’ with ‘trace’ as Mn, Co, Cr, Zn, Ni are essential elements for plants and animals.

Line 21: Be more specific than “some places”.

Line 24: What is “it”? Be specific.

Line 25: The final sentence should provide readers with a greater/broader take away message.

Introduction

Line 31: “Heavy metals” should no longer be used, instead, trace metals or toxic metals should me used.

Line 34: Metals do not under biodegradation. They can change oxidation state and bioavailability, but they themselves remain.

Line 39: Remove “are” before serve.

Line 63: what geologic features should be noted?

Line 67-71: This paragraph seems out of place. Maybe move it to the methods section.

Line 82: Add “focuses on” after “antimony-mercury,”

Line 88: Please add some expected hypotheses and mechanisms. Why would someone expect pollution in the area? What metals are expected to be enriched? Would the soil or sediment or water be more polluted?

Methods:

Line 92: Please state the countries and areas sampled. Right now, it looks like samples were from American Falls Idaho.

Figure 1: Tajikistan cannot be read on the figure. I suggest putting the text in black font in the upper right corner of the map. Also, the upper inset and bottom map cannot be read at all.

Line 102: Add “the” between “In” and “Laboratory”.

Line 107: correct spelling for “weighed”.

The methods INAA description was good.

Results

Line 172: Please describe citation #1 with some additional details.

Line 178: Remove “and their compounds”

Discussion

Figure 3: This is not needed. Samples with high sorption capacity have higher overall metal concentrations. So nothing surprising or noteworthy. Instead, showing a comparison of the unexposed and direct anthropogenic polluted soils mentioned in lines 202 to 209 would be much better and more interesting.

Line 239: Change “genetic type” as it is unclear about its meaning.

Conclusions

The attirbution of toxic metals to specifc processes need to be in the discussion with citations or references to substantiate claims of flood deposition, river transport, and mining sources. As described, the claims in the conclusions are not supported or articulated.

Author Response

Thanks for your comments and notes.
Answers to your remark in the attached file.

Reviewer 2 Report

The authors deal with the load of soils and sediments by potentially toxic elements in Zarafshon Valley (Tajikistan). The methodology uses standard tools for evaluation. The merit of the work is the ambition to describe the general state of valley influenced by natural and anthropogenic sources of soil and sediment load by high number of potentially risky elements. The authors correctly observe that intensive research will be needed in local areas with serious contamination by risky elements. The manuscript is well prepared nevertheless some changes or correction are needed.

Material and Methods - the description of study area is missing. Please use the chapters from Introduction describing the area of interest and move it to the Material and Methods as individual sub - chapter (2.1 Study area) and describe better the area of interest (including better general location, meteorological data, soils, geology, potential sources of contamination, etc.).

Results - Figure 2 (page 5) - no comment about Hg with high values….

References - there are 30% references in Russian language that are available for limited spectrum of potential readers. In my opinion it is tu many....

Some formal mistakes must be corrected: 

r. 83 - gold, antimon and mercury

r. 193 - can be better described (not describes)

r. 199 - can be explained by the (not be the)

r. 264 - ..by V, Cr, Mn, Co, Ni and Zn

r. 275 - ..V, Cr, Ni and Zn

r. 278 - For mercury... - use chemical symbols or full names of elements uniformly (better For Hg…).

Author Response

(The authors gave the same response as above.)

Round 2

Reviewer 1 Report

I thank the authors for making many changes to their manuscript. It is a treasure trove of geochemical data. However, the interpretations are very limited and cause the discussion to be vague and unhelpful for readers. To me, this vagueness would be helped considerably by address the two comments that were not sufficiently addressed from my first review.

1) My comment "It would be nice to have a figure or table in the manuscript that shows V, Cr, Mn, Co, Ni, Zn, As, Sb, Ba, W, Hg concentrations in soils and sediments separately, not just the Igeo values."

Author Response: Thank you for remarks. Metal and metalloid concentration data shown in Supplementary Materials. 

Reviewer new response: My point is that most readers will not see the supplementary materials and the concentration data should be shown or described in the manuscript to help provide context to the enrichment. It is not even written in the text. The Igeo values need the context. For example, if As concentrations are 0.1 ppb in the rock and 5 ppb in the soil, it would suggest a high enrichment while in reality, the concentrations are very small.

2) In my comment "Further, the authors should consider if metals that exist as cations (Co, Ni, Zn, Ba) or oxyanions (Cr, As, V, U) are present in soils and sediments as anions can have greater mobility as soil materials (organic matter, clays, oxides) typically are negatively charged."

Author Response: Thank you for remarks. This study focus on elemental content in soil and sediment in investigated area, using the devices and capabilities available to us. The study of the presence of anions or oxyanions are plan of future.

Reviewer new response: The discussion needs this information. As examples, Cr, As, V, and U whether you measured it or not, is almost certainly an oxyanionic form at the present environmental conditions described in this study. Thus, it will move faster through soils than cationic elements. I cannot recommend publication of your study unless this key information is added to the discussion.

Author Response

Dear Reviewer

Authors new response in attached file.

Thank you.

Reviewer 2 Report

The authors corrected the manuscript following my recommendations. I agree with manuscript publication.

Author Response

Thanks for your remarks and recommendations.

Round 3

Reviewer 1 Report

Thank you for adding the trace metal and metalloid concentration data to the manuscript. I believe this manuscript should be accepted in its current form.